# Measles and Rubella Incidence and Molecular Epidemiology in Senegal: Temporal and Regional Trends during Twelve Years of National Surveillance, 2010–2021

**DOI:** 10.3390/v14102273

**Published:** 2022-10-17

**Authors:** Mamadou Malado Jallow, Bacary Djilocalisse Sadio, Marie Pedapa Mendy, Sara Sy, Amary Fall, Davy Kiori, Ndiendé Koba Ndiaye, Yague Diaw, Déborah Goudiaby, Boly Diop, Mbayame Ndiaye Niang, Elizabeth J. A. Fitchett, Ndongo Dia

**Affiliations:** 1Département de Virologie, Institut Pasteur de Dakar, Dakar BP 220, Senegal; 2Division Surveillance Epidémiologique et Riposte Vaccinale du Ministère de la Santé et de L’action Sociale, Dakar BP 220, Senegal; 3Clinical Research Department, London School of Hygiene and Tropical Medicine, London WC1E 7HT, UK

**Keywords:** measles, rubella, Senegal, epidemiology, surveillance, genotype

## Abstract

We investigated the epidemiology of measles and rubella infections in Senegal based on data from twelve consecutive years of laboratory-based surveillance (2010–2021) and conducted phylogenetic analyses of circulating measles viruses. Sera from measles-suspected cases were collected and tested for measles and rubella-specific IgM antibodies using enzyme-linked immunosorbent assays (ELISA). Throat swabs were collected from patients with clinically diagnosed measles for confirmation by reverse-transcription polymerase chain reaction (RT-PCR) and viral genotyping. Among 8082 laboratory-tested specimens from measles-suspected cases, serological evidence of measles and rubella infection was confirmed in 1303/8082 (16.1%) and 465/6714 (6.9%), respectively. The incidence of rubella is now low—0.8 (95% CI 0.4–1.3) cases per million people in 2021—whereas progress towards measles pre-elimination targets (<1.0 case per million people per year) appears to have stalled; there were 10.8 (95% CI 9.3–12.5) cases per million people in 2021. Phylogenetic analyses revealed that all Senegalese measles strains belonged to genotype B3. The rubella virus sequence obtained in this study was consistent with genotype 1C. Our national surveillance data suggest that despite their low incidence both measles and rubella remain endemic in Senegal with a concerning stagnation in the decline of measles infections that represents a significant challenge to the goal of regional elimination.

## 1. Introduction

Measles is one of the most devastating infectious diseases in humans and it was annually responsible for over 6 million deaths during the pre-vaccination era [1]. However, there has been remarkable progress in reducing the mortality and morbidity burden from measles through measles vaccination, with an estimated 134,200 and 110,000 measles-related deaths globally in 2015 and 2017, respectively [2,3]. Measles is a highly contagious systemic viral infection caused by a single-stranded RNA virus of the genus Morbillivirus, within the family of *Paramyxoviridae* [4]. It is transmitted via respiratory secretions or droplets with typical symptoms of cough, coryza, conjunctivitis, fever and rash, occasionally leading to life-threatening pneumonia or encephalitis, as well as long-term deafness, blindness and immunosuppression [4,5,6]. There are 24 known measles virus genotypes, divided into eight clades from A to H, with genotypes B3 and D8 now the most common causes of outbreaks globally [4,5,6,7].

Rubella, another vaccine-preventable infection that can mimic the measles syndrome and is similarly transmitted, is caused by the rubella virus (RV)—an enveloped virus of the Rubivirus genus in the Matonaviridae family [8]. Rubella infection during pregnancy can result in death or congenital rubella syndrome (CRS) in the fetus, which may involve sensorineural hearing impairment, eye defects, congenital heart disease, microcephaly, developmental delay, bone alterations and damage to the liver and spleen [8,9,10]. Although the burden is not well documented in many countries, it is estimated that in 2010 more than 100,000 babies were born with CRS worldwide [8,10].

Measles and rubella infections are ongoing public health threats worldwide and are targeted by the World Health Organization (WHO) for elimination [9,11]. The WHO currently recommends two doses of measles-containing vaccine (MCV) for all children [11]. The MCV was first introduced through the Expanded Program on Immunization (EPI) for infants in Africa in the 1980s, augmented with mass vaccination campaigns [12]. In 2011, the WHO African region set a target for measles elimination for 2020, with the pre-elimination goals of high (≥95%) measles vaccine coverage and <1.0 case per million people per year in all African countries [13]. Yet, progress has been hindered by a surge in cases globally between 2017–2019, with fears of imminent exacerbation secondary to vaccine program disruption caused by the COVID-19 pandemic [11].

A key pillar of the elimination goal is to establish robust surveillance systems across the region; the definition of elimination requires the absence of case detection in the presence of reliable surveillance [11,13,14]. In countries with high measles vaccine coverage, the WHO recommends switching to the combined measles-rubella vaccine; in parallel, rubella surveillance has also increasingly been incorporated into measles surveillance programs given the overlapping clinical presentations of the two infections [11]. 

Prior to 2014, children in Senegal were routinely immunized with one dose of measles-containing vaccine (MCV1) at 9 months of age through the national EPI, with coverage increasing from 57% in 2004 to 84% in 2013 [15]. In addition to routine immunization services, nationwide vaccination campaigns were conducted in 2006 and 2010 using measles-only vaccine (target age group: 9–59 months) and in 2013 using the combined measles-rubella vaccine (MRV) (target age group: 9 months–14 years), with 102% administrative coverage of the 2013 campaign [15]. In 2014, the MRV replaced the measles-only vaccine in the EPI in a two-dose schedule at 9 and 15 months, with 87% coverage in 2021 [15,16]. To complement these strategies, Senegal has had laboratory-based surveillance of measles established since 2000 [17], including molecular epidemiological investigations to support global elimination. This facilitated the detection of a major measles outbreak during 2009–2010 that affected people across the African continent [10,17,18]. Since the implementation of the combined MRV, serological rubella surveillance has been integrated into this program, through testing all measles-negative and indeterminate samples from suspected measles cases and active investigation of any potential outbreaks.

The *WHO Measles and Rubella Strategic Framework 2021–2030* has set the stage for renewed measles and rubella elimination targets in Africa following the significant setbacks before and during the COVID-19 pandemic [11]. Here, we describe an updated overview of the epidemiology, temporal trends and regional distribution of measles and rubella infections in Senegal during twelve consecutive years of national reference laboratory-based surveillance (2010–2021), including the molecular characterization of measles and rubella strains to determine circulating genotypes.

## 2. Materials and Methods

### 2.1. Case Definitions and Clinical Specimens

This retrospective analysis describes national data on all serological measles and rubella detections in samples collected from suspected measles cases throughout the fourteen regions of Senegal from January 2010 to December 2021, and nasopharyngeal swabs for molecular analyses from 2019 onwards. Case-finding for both infections is based on the WHO measles case definition: any person with fever and maculopapular (non-vesicular) generalized rash and cough, coryza or conjunctivitis (red eyes) or any person in whom a clinician suspects measles [19]. On identifying a suspected measles case, clinicians complete a case-based form in duplicate and collect a blood specimen which is sent to the WHO National Reference Laboratory for Measles and Rubella hosted at the Institut Pasteur de Dakar (IPD). All blood samples are collected within 30 days of the rash onset, transported to the laboratory at a controlled temperature (4 °C) and tested immediately upon arrival for serological evidence of measles, or subsequently, rubella infection. Following confirmed case detections (usually three or more positive samples from the same district) an outbreak investigation team conducts active surveillance of close contacts to offer vaccination, with some additional sampling conducted as further cases are identified (although in large outbreaks, vaccination is prioritized over sampling all suspected cases). Since 2019, the national surveillance protocol has recommended that a concurrent nasopharyngeal swab for all clinically suspected measles cases should be sent in viral transport medium for RT-PCR confirmation and genotyping, where possible. 

### 2.2. Laboratory Confirmation of Measles and Rubella Cases

Serum specimens are tested for measles and rubella specific immunoglobulin M (IgM) antibodies using WHO-recommended ELISA kits (Enzygnost Anti-Measles-Virus/IgM from Dade Behring or Siemens, Erlangen, Germany) according to the supplier’s protocols. Given the similar clinical presentations of measles and rubella and the WHO/AFRO measles surveillance guidance to exclude measles cases from rubella screening, samples are first tested for measles IgM antibody, with measles-negative and equivocal samples subsequently tested for rubella specific IgM antibody [19]. For both measles and rubella, a laboratory-confirmed case is defined by an IgM absorbance value difference (ΔA) of more than 0.2. An equivocal case is defined by an absorbance value difference between 0.1 and 0.2; all equivocal cases are re-tested for confirmation. 

### 2.3. Detection of Measles and Rubella Viruses by RT-PCR

All available nasopharyngeal swabs are tested by RT-PCR for the presence of measles, and then for rubella RNA only if negative for measles, independent of any serological results. Viral RNA is extracted from 200 µL of throat swab sample medium using the QIAamp Viral RNA kit (QIAGEN, Valencia, CA, USA) according to the supplier’s protocol. Extracted RNA is eluted in 60 μL of elution buffer and tested for the presence of measles or rubella RNA by real-time RT-PCR as per WHO standard protocols [20]. Tests are performed in singleplex using the specific primers and probe for each virus. The TaqMan Universal PCR Master Mix kit (Applied Biosystems, Waltham, MA, USA) is used for real time amplifications, as previously reported [21].

### 2.4. Measles and Rubella Genotyping

All RT-PCR measles or rubella positive nasopharyngeal samples are genotyped. After an initial step of cDNA synthesis using the RevertAid First Strand cDNA Synthesis Kit (Thermo Scientific, Vilnius, Lithuania), a nested PCR is performed to amplify targeted fragments for both measles and rubella viruses. For measles virus, PCR is carried out using primers previously described [17]. The GoTaq^®^ DNA Polymerase (Promega, Madison, WI, USA) is used for amplifications. The first PCR is carried out in a total reaction volume of 27.25 μL containing 12 μL of nuclease free water, 5 μL of 5X GoTaq^®^ Reaction Buffer, 2 μL of MgCl2, 1 μL of dNTP (10 mM), 1 μL of forward primer (20 μM), 1 μL of reverse primer (20 μM), 0.25 μL of GoTaq^®^ DNA Polymerase (5 u/µL) and 5 μL of cDNA template. The reaction mixture is amplified in a thermocycler under the following conditions: an initial denaturation step of 3 min at 95 °C followed by 35 PCR cycles at 95 °C for 30 s, 55 °C for 30 s, and 72 °C for 1 min followed by a final step at 72 °C for 5 min. The Nested PCR is performed on the resulting diluted (1:50) amplicon using the same condition as the first PCR. Regarding amplification of the rubella virus, the reaction mixture of the first PCR contains 5 μL of cDNA template, 5 μL of 5X GoTaq^®^ Reaction Buffer, 1 μL of RubF1 (5′-CCCACCGACACCGTGATGA-3′) forward primer (20 μM), 1 μL of RubR1 (5′-CCAGGTCTGCCGGGTCTC-3′) reverse primer (20 μM), 2 μL of MgCl2, 1 μL of dNTP (10 mM), 0.25 μL of GoTaq^®^ DNA Polymerase (5 u/µL) and 12 μL nuclease free water and the cycling conditions are 3 min at 95 °C followed by 35 cycles at 95 °C for 30 s, 55 °C for 30 s, and 72 °C for 1 min, with a final step at 72 °C for 5 min. For the nested PCR, internal primers RubF2 (5′-GTGATGAGCGTGTTCGCCC-3′) and RubR2 (5′-GCDGTGGTGTGTGTGCC-3′), and 1 μL of the first PCR product are used. Amplification products are analyzed in a 1.5 % agarose gel stained with ethidium bromide, using 1xTAE as the electrophoresis running buffer. For measles, the 450-nucleotide region in the N gene (nucleotide positions (nts) 1233–1682) is sequenced, as previously described [17]. This region recommended by WHO for genotyping measles virus encodes the C-terminal end of the nucleoprotein. With regard to rubella virus, an 800 nt fragment of the E1 coding region containing the 739 nt World Health Organization (WHO) recommended sequence window (nts 8731–9469) is amplified and sequenced. PCR products are initially purified with the NucleoSpin^®^ Gel and PCR Clean-Up (Macherey-Nagel GmbH & Co. KG, Düren, Germany) according to the supplier’s protocol, and then sent to Genewiz (Essex, United Kingdom) for Sanger sequencing in both forward and reverse directions. 

### 2.5. Phylogenetic Analysis of Measles Strains

The sequences obtained in Fasta format are initially cleaned as required using the GeneStudio software (GeneStudio™ Pro, version: 2.2.0.0, 8 November 2011) and then aligned with related measles virus (MeV) reference strains as designated by WHO using the ClustalW alignment program within the BioEdit software [22]. Reference strains are retrieved from GenBank and Measles Nucleotide Surveillance (MeaNS) online databases. Phylogenetic trees are constructed by the neighbor-joining method based on the Kimura two-parameter model in MEGA version 7 [23]. The reliability and robustness of the branching orders are analyzed by bootstrap analysis of 1000 replicates. Only bootstrap replicates with values ≥70 are shown on the tree. Identical strains are not shown to better illustrate the range of variants in Senegal. 

### 2.6. Data Management and Statistical Analysis 

The laboratory results together with the case investigation data are entered into the measles case-based surveillance database using the Epi Info™ software. Anonymized, case-based surveillance data are shared by IPD with the Ministry of Health and WHO on a weekly basis. The data presented in this study were exported into Microsoft Excel; they were cleaned and analyzed using R (3.0.1) and Stata/SE (17.0). National incidence rates of laboratory-confirmed measles and rubella were estimated per million people (all ages) per year, based on IgM-positive detections (i.e., not including cases with equivocal levels of IgM on repeated testing); confidence intervals were calculated using the Poisson distribution. Denominator population data were sourced from L’Agence Nationale de la Statistique et de la Démographie (ANSD) (https://senegal.opendataforafrica.org, accessed on 20 September 2022). For regional incidence estimates of incidence during the whole study period (2010–2021) the median population sizes per region between 2010 and 2021 were used.

### 2.7. Ethical Considerations

The combined surveillance protocol was adapted and finalized by IPD, supported by WHO in the context of the measles elimination program; it was approved by the Senegalese National Ethical Committee of the Ministry of Health. Anonymized data are collected as part of routine surveillance and the need for written consent has been waived. Samples are collected by local health care workers as part of clinical care with verbal consent.

## 3. Results

### 3.1. Suspected and Confirmed Cases

Overall, during 2010–2012, 8142 serum samples of suspected cases were reported from all cities and districts of Senegal. In total, 60 and 65 eligible samples were not tested for measles and rubella, respectively, due to shortage of measles/rubella kits, inadequate sample volumes or lack of proper identification. Of the 8082 specimens tested, 1303 (16.1%) were measles IgM-positive; 179 (2.2%) yielded equivocal results (Table 1). Of the 6714 measles-negative and equivocal samples tested for the presence of rubella specific IgM antibodies, 465 (6.9%) were positive and 214 (3.2%) were classified as equivocal (Table 1). Throat swab samples were collected from 71 clinically suspected cases in 2019 and 2020, of which 62 (87.3%) were positive for measles virus; only one sample (1.4%) was found to be positive for rubella. 

### 3.2. Sociodemographic Characteristics

The sex, age and measles vaccination status of patients with suspected, confirmed and equivocal measles or rubella are reported in Table 1. The median ages for suspected (all tested), measles (IgM-positive) and rubella (IgM-positive) cases were 5 (IQR 2–8), 4 (IQR 2–8) and 5 (IQR 3–8) years, respectively. Only 10 (10.1%) and 7 (8.6%) IgM-positive rubella cases occurring from 2015 and 2016 onwards, respectively, (i.e., following introduction of the combined MRV) had a documented measles vaccination history, two of whom were >10 years old. 

### 3.3. Temporal Trends and Incidence of Confirmed Measles and Rubella

The annual numbers of suspected cases, positive and equivocal results and the incidence for measles and rubella infections during the surveillance period (2010–2021) are shown in Table 2. The monthly (seasonal) distribution of positive and negative cases of measles and rubella are depicted in Figure 1 and Figure 2, respectively.

During this surveillance period, three seasonal epidemics of measles were detected (2009–2010, 2015–2016 and 2019–2020) (Figure 1). In 2018, measles detections were at the lowest level recorded in the study period, with only seven IgM-positive cases. However, since 2019, there has been an overall increase in the number of measles detections. The overall national incidence of laboratory-confirmed measles during the study period was 7.3 (95% CI 6.9–7.7) cases per million people/year (Table 2); there was no sustained decrease in incidence over time (6.0 (95% CI 5.5–6.5) and 8.5 (95% CI 7.9–9.1) cases per million people/year during the 2010–2015 and 2016–2021 time periods, respectively).

The highest number of rubella IgM-positive cases were recorded in 2011 and 2014, but with decreases observed from the year 2015 to only 13 IgM-positive cases in 2021. The overall national incidence of rubella during the surveillance period was 2.6 (95% CI 2.4–2.9) cases per million people/year (Table 2). There were 4.8 (95% CI 4.3–5.3) and 0.8 (95% CI 0.6–1.0) cases per million people/year during 2010–2015 and 2016–2021, respectively.

The monthly distribution of measles confirmed cases showed a clear seasonal pattern during the three epidemic seasons—gradually increasing from January with a consistent detection peak in March (Figure 1). Rubella transmission appeared to increase from February to June in the first five years of the study period (2010–2014) (Figure 2).

### 3.4. Geographical Distribution

The number of suspected and confirmed cases of measles and rubella and their incidence by region are shown in Table 2. Figure 3 illustrates the regional distribution of measles and rubella incidence comparing the time periods 2010–2013, 2014–2017 and 2018–2021. The largest number of serum samples were received from the capital city, Dakar (1952; 24.2%), which was also the region representing the largest proportion of laboratory-confirmed cases (508; 39.0%). In fact, two-thirds (873; 67%) of laboratory-confirmed cases were detected in just four regions (Dakar, Kédougou, Tambacounda and Thies), with no confirmed cases in the southern Sedhiou region (Table 2). However, the incidence of laboratory-confirmed measles has been consistently highest in the Kédougou region, most recently with an increase to 118.6 (95% CI 95.2–145.9) cases per million people/year between 2018–2021 (Figure 3), compared to 13.0 (11.9–14.1) cases per million people/year in Dakar.

Regarding rubella infections, most IgM-positive cases (140/465; 30.1%) were from Thies, a region adjacent to the capital city Dakar, followed by Dakar (97/465; 20.9%), Fatick (36/465; 7.7%) and Tambacounda (28/465; 6.0%). As with measles, however, the highest incidence was in Kédougou (7.1 (95% CI 3.9–12.0) cases per million people/year) (Table 2; Figure 3).

### 3.5. Measles Virus Genotyping and Phylogenetic Analysis

All 62 measles-positive samples by RT-PCR were successfully genotyped—55 from 2019 and 7 from 2020. To compare with previously reported strains, all 62 sequences were analyzed using the nucleotide Basic Local Alignment Search Tool (BLASTn) and the results indicated similarities with genotype B3 strains. The 450 nucleotide sequences of MeVs obtained in this study were further compared with the WHO reference sequences recommended for genotype identification; the phylogenetic tree confirmed that Senegalese measles viruses belonged to genotype B3 (Figure 4). Furthermore, the B3 strains identified in this study were compared with sequences of other B3 sub-genotypes retrieved from the GenBank database, with the phylogenetic tree indicating that all the Senegalese B3 strains belonged to the B3.1 sub-genotype (Appendix A). When compared with WHO-recommended reference strains, the rubella E1 gene nucleotide sequence obtained in this study was consistent with genotype 1C.

## 4. Discussion

The Republic of Senegal has relatively high measles vaccine coverage and is one of only twelve African countries meeting measles surveillance targets [24,25]. It is, therefore, a key setting in West Africa in which to describe up-to-date temporal and regional patterns in measles and rubella epidemiology—including the molecular characteristics of circulating strains—in order to inform understanding of the challenges in achieving elimination and effective surveillance.

Our data captures trends during more than a decade of measles elimination efforts in the African region. Indeed, the large decrease in measles incidence between 2010 and 2011 may reflect the success of the 2010 vaccination campaign; the effect of the 2013 campaign on measles control is less apparent, albeit starting from a much lower baseline incidence. As previously documented, our estimate of overall measles incidence in Senegal between 2010–2021 (7.3 (95% CI 6.9–7.7) cases per million people per year) is relatively low compared to other West African countries, although significantly higher than the pre-elimination of <1.0 per million people per year, set in 2011 [13,25]. However, this figure masks the annual variability in incidence. Most concerningly, there have consistently been more than 10 cases per million people annually for the past three years—with incidence previously 1.0 (95% CI 0.5–1.6) and 0.4 (95% CI 0.2–0.9) per million people per year in 2017 and 2018, respectively. This is despite stepping up to a two-dose vaccine schedule in 2014 with high population coverage and reflects similar setbacks experienced in measles control documented globally during this time period. Importantly, we did not have complete information on residential status or travel history, meaning we cannot assess the degree to which imported cases and outbreaks in surrounding countries are driving the observed incidence in Senegal.

There has been a sustained decline in confirmed rubella infections to a mean incidence of 0.8 (95% CI 0.6–1.0) between 2016 and 2021, following the introduction of the combined measles-rubella vaccine, mirroring the journey towards elimination with immunization of several southern hemisphere countries, including in the WHO Region of the Americas [26,27]. Despite the high reported coverage of the 2013 combined MRV campaign, it is interesting that there was a higher number of rubella cases in 2014 than in 2013, with the subsequent decline appearing to occur after the introduction of the MRV into the routine schedule that year. It is important to consider that rubella cases were detected through a surveillance system designed to detect measles, and clinical presentations of rubella may not meet suspected measles case definitions; 20–50% of rubella infections do not include a rash [28]. As in other settings, the incorporation of rubella testing into the Senegalese measles surveillance program has facilitated a pragmatic and cost-efficient means of monitoring rubella infections [18,19,29]. However, sufficient surveillance to support rubella elimination in future would require additional active monitoring of potential rubella outbreaks, particularly in girls and women of reproductive age, and including clinical review of potential CRS cases [11].

Consistent with other studies in Burkina Faso, Italy and Pakistan [30,31,32], the majority of both measles (96.9%) and rubella (77.9%) infections were in unvaccinated patients (or those with unknown vaccination status). The type of vaccine (i.e., MCV vs. MRV), however, is not captured in our data—and so inferences cannot be made regarding rubella vaccine failure, although very few rubella cases occurred in children with a documented vaccine history following MRV introduction in 2014. Importantly, we have observed that both measles and rubella remain early childhood diseases in Senegal despite the immunization program. Most confirmed measles infections were in children less than five years of age (55.8%), suggesting coverage within the EPI may still be insufficient. Conversely, rubella cases were more commonly reported in the 5–15 years age group, in line with findings in Pakistan and Cameroon, where 63% and 97% of laboratory confirmed cases belonged to this age group, respectively, suggesting the need for additional catch-up immunization [29,32]. This is of particular concern given that 21.5% of women in Senegal give birth before the age of 18 (32.4% in rural districts), heightening the risk of congenital rubella as a complication in this age group [33].

Our reported male–female ratio of 1:1.2 in IgM-positive measles cases is consistent with findings from Zaidi et al. [32], and disproportionate to the child (<7years) sex ratio of 1:1.06 in Senegal (www.unfpa.org, accessed on 1 August 2022). The even higher male–female ratio of suspected cases of 1:1.4 may suggest a difference in care-seeking or other factors influencing case-finding, as sex-related differences in measles incidence are not consistently reported elsewhere [34,35].

Our data demonstrate that measles and rubella outbreaks in Senegal appear to have a seasonal pattern coinciding with the end of dry season, which is consistent with data throughout Africa and Asia [6,30,35,36,37,38] with some exceptions, including Zimbabwe and Uganda [18,38]. Apparent outbreaks of measles in Senegal were noted in 2010, 2016, 2019 and 2020 with most detections between January through April (peaking in March). Regarding rubella, most infections were noted from 2010 to 2014 with peaks between February through June each year. Unlike the confirmed measles outbreaks, the consistent annual seasonal pattern of IgM-negative suspected measles cases is notable. The discrepancy between the number of suspected and confirmed cases likely reflects other etiologies causing measles-like rashes in Senegal, including common respiratory viral exanthems. The case-finding criteria are broad to maintain surveillance sensitivity and atypical cutaneous rashes may also be included. In a previous study it was shown that among 3358 measles suspected patients collected between 2012 and 2017, 52.5% were attributable to human herpes viruses, with varicella zoster virus (VZV) being the most commonly detected (44.3%), whereas only 6.7% and 4.5% were positive for measles and rubella IgM, respectively [21].

Geographically, both measles and rubella infections were unevenly distributed and clustered around the population-dense urban regions, as has been previously documented in Zimbabwe [18]. In harder-to-reach rural regions, such as Kédougou, with much lower population densities, we have demonstrated pockets of relatively high measles and rubella incidence, which are obscured in national estimates. This highlights the critical role of surveillance programs within elimination strategies to report regional and district-level incidence in real-time, in order to identify areas with incomplete vaccination coverage and potential outbreaks requiring supplementary immunization activities (SIA).

An important limitation of the interpretation of our results is the reliance on case-finding by healthcare workers, and that cases may be missed in the community who do not present to healthcare. There are number of potential barriers to healthcare access, and temporal trends may be difficult to interpret given possible time-related disruptions to healthcare provision. For example, in 2020 and 2021, the possible impact of the COVID-19 pandemic on the occurrence, identification and reporting of suspected and confirmed cases is unclear. However, other than the COVID-19 pandemic, we are not aware of national disruptions to healthcare provision and our surveillance system was operational throughout the study period, with any local disruptions to healthcare provision unlikely to affect the overall interpretation of our results.

Our phylogenetic analyses supported evidence from one previous study [17] that there is exclusive circulation of the B3 measles genotype in Senegal, as in several other sub-Saharan African [39,40], European [7,31,41,42] and Asian [32] countries. This differs to some settings in Europe, including France Italy and Romania where co-circulation of genotypes D8 and B3 have been detected [7,43,44]. All B3 strains detected in the current study clustered into a B3.1 sub-genotype—previously detected in Cameroon, Central African Republic, Ghana, Nigeria, Kenya, Tanzania and Pakistan [32,40,45,46,47]. However, the B3.3 lineage previously reported to be the dominant circulating sub-genotype in the period 2009–2011 in Senegal [17] was not detected in the present study, possibly due to a non-exhaustive sequencing. The single Rubella sequence obtained in this study belongs to the genotype 1C, a genotype which circulated in Brazil, Peru and the West Coast of South America [48]. Further molecular analysis of rubella viruses in Senegal will be challenged by the ongoing progress towards elimination, although it may play an important role in the containment and surveillance of future outbreaks.

## 5. Conclusions

Our national surveillance data highlight stalled progress in achieving the elimination of measles and should inform the key challenges to be addressed in the renewed 2030 elimination strategy, including regional heterogeneity in measles incidence, seasonal outbreaks, and the coverage of both routine immunization and SIAs. Children younger than 5 years and those aged 5–15 years old remain the most vulnerable groups to measles and rubella infection, respectively, representing key target groups for catch up immunization efforts. High-quality surveillance remains a critical pillar of the elimination strategy going forward, especially given the relatively high baseline of infections in 2019 and 2020 and the threat of new outbreaks following potential COVID-19 pandemic-related vaccination disruption. We highlight the endemic circulation of the sub-Saharan genotype (B3), the presence of a genotype 1C Rubella strain and the need for continued molecular surveillance in the event of new outbreaks. The decline in rubella incidence has demonstrated the success of the combined MRV immunization program; however, rubella incidence is underestimated without a rubella-specific case definition and by testing only measles-negative cases. Maintaining immunization coverage above the immunity threshold, supporting high-quality district-level surveillance and implementing prompt control measures around imported cases [49,50] will be critical to achieving long-term elimination of both viruses for Senegal and the West Africa region.

## Figures and Tables

**Figure 1 viruses-14-02273-f001:**
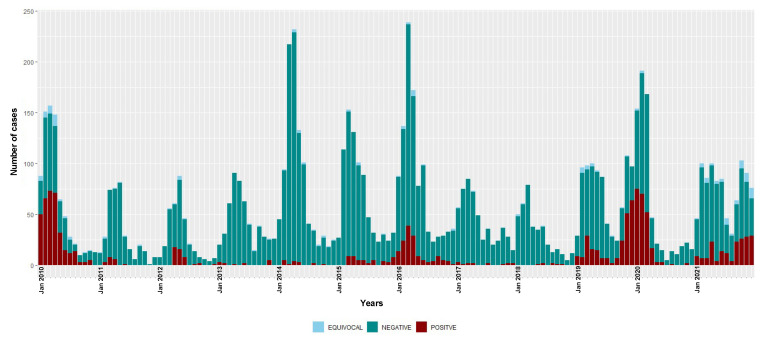
Temporal distribution of measles IgM-positive, equivocal and negative cases, 2010–2021. Bars are stacked and total bar height represents the number of suspected cases tested for measles IgM.

**Figure 2 viruses-14-02273-f002:**
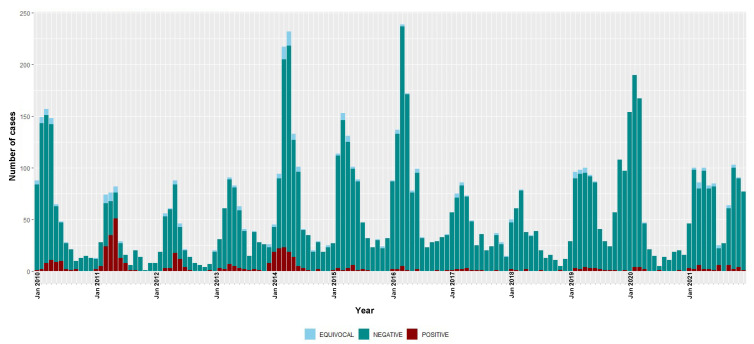
Temporal distribution of rubella IgM-positive, equivocal and negative cases, 2010–2021. Bars are stacked and total bar height represents the number of suspected cases tested for rubella IgM.

**Figure 3 viruses-14-02273-f003:**
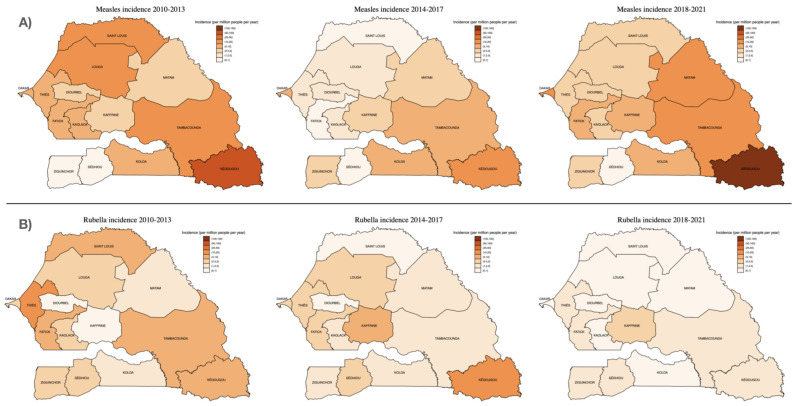
Incidence of (**A**) measles and (**B**) rubella IgM detections by region, over three consecutive surveillance periods: 2010–2013, 2014–2017 and 2018–2021. Incidence estimates were calculated using the median population size for each region and time period (population data sourced from Agence Nationale de la Statistique et de la Démographie (ANSD): www.ansd.sn, accessed on 1 August 2022).

**Figure 4 viruses-14-02273-f004:**
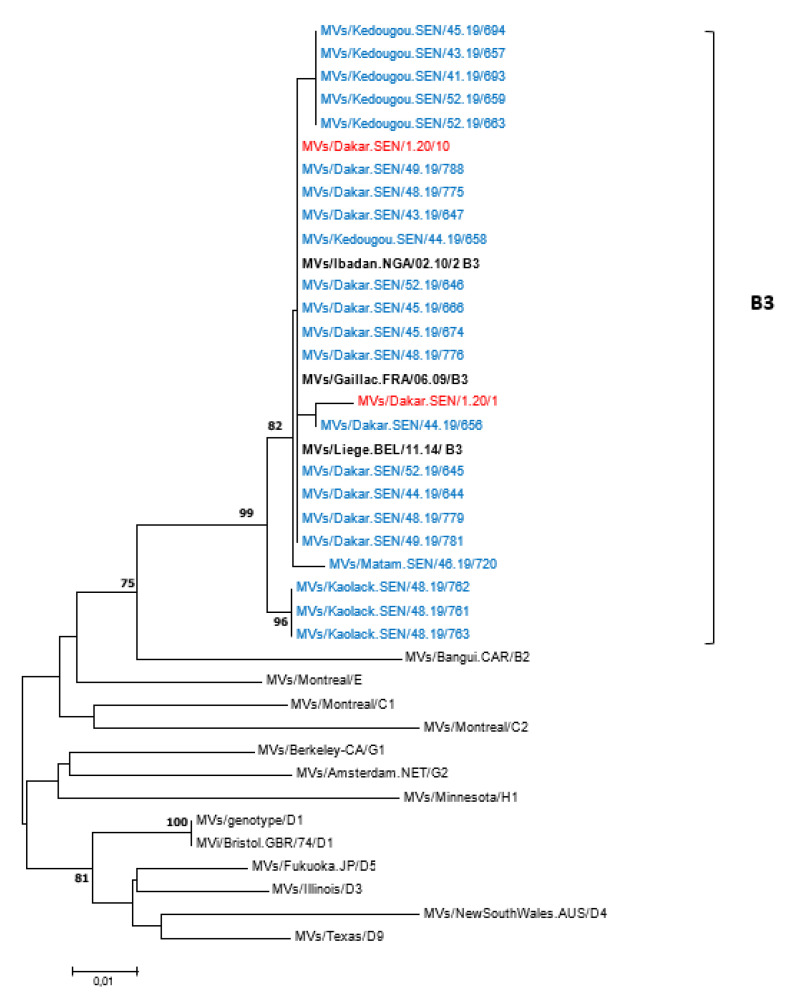
Phylogenetic relationships of measles strains detected in Senegal (2019 and 2020) with WHO reference sequences. The tree was constructed based on the 450 nucleotides coding the C-terminal end of the nucleoprotein gene using the neighbor-joining method under the Kimura 2-parameter model, as implemented in MEGA 7 software. The robustness of the nodes was tested with 1000 bootstrap replications and bootstrap support values greater than 70 are shown at the nodes. Senegalese strains are denoted in blue (2019 sequences) or red (2020 sequences) diamond. Scale bar indicates nuclelotide substitutions per site. Identical strains are not shown.

**Table 1 viruses-14-02273-t001:** Sex, age and vaccination status of suspected, IgM-positive and equivocal cases of measles and rubella in Senegal, 2010–2021.

	Suspected Cases	Measles (*n* = 8082)		Rubella (*n* = 6714) ^a^	
		IgM-Positive	Equivocal	IgM-Positive	Equivocal
	*n* (%)	*n* (%)	*n* (%)	*n* (%)	*n* (%)
Sex					
Male	4560 (57.0)	744 (57.1)	105 (58.7)	277 (59.6)	108 (51.4)
Female	3439 (43.0)	559 (42.9)	74 (41.3)	188 (40.4)	102 (48.6)
Missing	83 (1.0)	0 (0.0)	0 (0.0)	0 (0.0)	4 (1.9)
Age-group (years)					
[0 to <2]	1363 (17.2)	272 (20.9)	31 (17.3)	48 (10.3)	25 (11.7)
[2 to <5]	2444 (30.1)	462 (35.5)	57 (31.8)	151 (32.5)	32 (15.0)
[5 to <15]	3502 (44.2)	387 (29.7)	68 (38.0)	244 (52.5)	140 (65.4)
[15 to <50]	600 (7.6)	153 (11.7)	22 (12.3)	13 (2.8)	11 (5.1)
50+	23 (0.3)	1 (0.1)	0 (0.0)	0 (0.0)	1 (0.5)
Missing	150 (1.9)	28 (2.2)	1 (0.6)	9 (1.9)	5 (2.3)
Measles vaccination status					
Vaccinated ^b^	968 (12.0)	35 (2.7)	17 (9.5)	103 (22.2)	34 (15.9)
Unvaccinated or unknown	7114 (88.0)	1268 (97.3)	162 (90.5)	362 (77.9)	180 (84.1)
Total/tested (%)	8082 (100)	1303/8082 (16.1)	179/8082 (2.2)	465/6714 (6.9)	214/6714 (3.2)

^a^ Measles-negative or indeterminate samples only. In total, 65 measles-negative or indeterminate samples were not tested for rubella, due to insufficient sample volumes or laboratory supplies shortages. ^b^ Has received at least one dose of the measles vaccine, or the combined measles-rubella vaccine, which was introduced into the routine schedule from 2014.

**Table 2 viruses-14-02273-t002:** The number and vaccination status of tested suspected cases, and the number and incidence of confirmed measles and rubella cases by year and region.

	Tested Suspected Cases	Measles Vaccination Status ^a^	Measles ^b^			Rubella ^b^		
		At Least One Documented Vaccination	IgM- Positive	Equivocal	Incidence per Million People per Year	IgM- Positive	Equivocal	Incidence per Million People per Year
	*n*	*n* (%) ^c^	*n* (%) ^c^	*n* (%) ^c^	(95% CI) ^d^	*n* (%) ^c^	*n* (%) ^c^	(95% CI) ^d^
Year (all regions)								
2010	731	76 (10.4)	344 (47.1)	40 (5.5)	27.1 (24.3–30.2)	45 (12.8)	26 (6.3)	3.6 (2.6–4.8)
2011	366	85 (23.2)	18 (4.9)	7 (1.9)	1.4 (0.8–2.2)	140 (40.2)	25 (7.2)	10.7 (9.0–12.7)
2012	338	41 (12.1)	46 (13.6)	8 (2.4)	3.4 (2.5–4.6)	42 (14.4)	12 (4.1)	3.1 (2.3–4.2)
2013	525	86 (16.4)	13 (2.5)	4 (0.8)	0.9 (0.5–1.6)	34 (6.6)	15 (2.9)	2.5 (1.7–3.5)
2014	989	165 (16.6)	16 (1.6)	15 (1.5)	1.1 (0.6–1.8)	105 (10.8)	48 (4.9)	7.4 (6.1–9.0)
2015	804	123 (15.3)	50 (6.2)	6 (0.7)	3.4 (2.6–4.5)	18 (2.4)	22 (2.9)	1.2 (0.7–2.0)
2016	994	163 (16.4)	146 (14.7)	15 (1.5)	9.7 (8.2–11.5)	12 (1.4)	16 (1.9)	0.8 (0.4–1.4)
2017	510	81 (15.9)	15 (2.9)	2 (0.4)	1.0 (0.5–1.6)	11 (2.2)	14 (2.7)	0.7 (0.4–1.3)
2018	377	37 (9.8)	7 (1.9)	4 (1.1)	0.4 (0.2–0.9)	7 (1.9)	5 (1.3)	0.4 (0.2–0.9)
2019	856	94 (11.0)	239 (27.9)	16 (1.9)	14.7 (12.9–16.6)	20 (3.2)	17 (2.7)	1.2 (0.8–1.9)
2020	682	17 (2.5)	223 (32.7)	5 (0.7)	13.3 (11.6–15.2)	10 (2.2)	1 (0.2)	0.6 (0.3–1.1)
2021	911	0 (0.0)	186 (20.4)	57 (6.3)	10.8 (9.3–12.5)	13 (5.8)	8 (3.5)	0.8 (0.4–1.3)
Region ^e^ (2010–2021)								
Dakar	1952	272 (13.9)	508 (26.0)	46 (2.4)	13.0 (11.9–14.1)	97 (6.8)	32 (2.2)	2.5 (2.0–3.0)
Diourbel	368	27 (7.3)	46 (12.5)	7 (1.9)	2.3 (1.7–3.1)	9 (2.9)	4 (1.3)	0.5 (0.2–0.9)
Fatick	467	35 (7.5)	48 (10.3)	7 (1.5)	4.8 (3.6–6.4)	36 (8.6)	10 (2.4)	3.6 (2.5–5.0)
Kaffrine	547	48 (8.8)	31 (5.7)	10 (1.8)	3.9 (2.7–5.6)	19 (3.7)	18 (3.5)	2.4 (1.5–3.8)
Kaolack	448	47 (10.5)	39 (8.7)	9 (2.0)	3.2 (2.3–4.4)	19 (4.7)	11 (2.7)	1.6 (0.9–2.4)
Kédougou	455	30 (6.6)	139 (30.6)	15 (3.3)	70.9 (59.6–83.7)	14 (4.5)	17 (5.5)	7.1 (3.9–12.0)
Kolda	269	43 (16.0)	59 (21.9)	4 (1.5)	6.8 (5.2–8.7)	9 (4.3)	5 (2.4)	1.0 (0.5–2.0)
Louga	438	71 (16.2)	63 (14.4)	8 (1.8)	5.4 (4.1–6.9)	25 (6.8)	19 (5.1)	2.1 (1.4–3.2)
Matam	330	18 (5.5)	72 (21.8)	3 (0.9)	9.2 (7.2–11.6)	9 (3.5)	12 (4.7)	1.1 (0.5–2.2)
Sédhiou	335	26 (7.8)	0 (0.0)	5 (1.5)	0.0 (0.0–0.6)	13 (3.9)	12 (3.6)	2.1 (1.1–3.6)
St. Louis	323	28 (8.7)	51 (15.8)	12 (3.7)	4.2 (3.1–5.6)	24 (8.9)	11 (4.1)	2.0 (1.3–3.0)
Tambacounda	658	37 (5.6)	119 (18.1)	24 (3.7)	5.6 (3.8–7.9)	28 (5.3)	14 (2.7)	3.0 (2.0–4.3)
Thiès	1056	201 (19.0)	107 (10.1)	22 (2.1)	4.6 (3.8–5.6)	140 (14.9)	32 (3.4)	6.0 (5.1–7.1)
Ziguinchor	431	85 (19.7)	18 (4.2)	7 (1.6)	2.1 (1.2–3.3)	21 (5.1)	17 (4.1)	2.5 (1.5–3.8)
Total	8082	968 (12.0)	1303 (16.1)	179 (2.2)	7.3 (6.9–7.7)	465 (6.9)	214 (3.1)	2.6 (2.4–2.8)

^a^ Measles-only vaccine prior to 2014, replaced by the measles-rubella combined vaccine nationally from 2014. ^b^ 8082 suspected cases were tested for measles IgM and 6714 measles-negative or measles-equivocal suspected cases were tested for rubella IgM. ^c^ Denominators for proportions are the number of suspected cases who were tested for measles or rubella per year or region. ^d^ Incidence estimates reflect the number of confirmed (IgM positive cases) detected in all regions using annual population denominators sourced from Agence Nationale de la Statistique et de la Démographie (ANSD) (https://senegal.opendataforafrica.org, accessed on 1 August 2022). Despite active surveillance following positive case detection, incidence rates are likely to be underestimated for both measles and rubella as not all suspected cases undergo sampling during larger outbreaks. ^e^ Data missing for region of 5 tested suspected cases.

## Data Availability

The datasets supporting the conclusions of this article are included within the article and its tables and figures. Additional data may be available from the corresponding author upon reasonable request.

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
