# Peer review of "Measles and Rubella Incidence and Molecular Epidemiology in Senegal: Temporal and Regional Trends during Twelve Years of National Surveillance, 2010–2021"

_viruses, 2022, doi:10.3390/v14102273_

Round 1

Reviewer 1 Report

The article under review presents long-term measles surveillance data from Senegal. While the information is of interest, not everything is clear and some improvements for data presentation and interpretation can be made. As general comments, Introduction and Discussion could be more concise and focussed and the number of tables and figures could be reduced. Also tenses in the Methods part should not be mixed to describe what has been done. In the following, several comments and suggestions are provided, referring if not otherwise stated to lines in the PDF document.

19: why were not all samples tested? Maybe leave the number of received specimens out from the Abstract in order not to confuse the readers

24/25: please use named strains instead

Abstract conclusion differs from what is said earlier: rubella declined

33: rubella is far less contagious than measles

53/54: please revise this part of the sentence: measles is not a threat to mortality and morbidity, but the cause of this

57: please update virus classification

59/60: eye defects missing

69: was rubella vaccination introduced with the MR campaign in 2013 or what is the situation for RCV?

80: is there any rubella-specific surveillance in Senegal?

89-91: is there a specific rubella case definition as well?

96-98: does the outbreak investigation team follow up on single confirmed cases or only if there are more than one (if yes, how many?)?

101-103: what determines whether a swab is collected (how is this subset of patients determined?)?

111-114: please rephrase for clarity – at the moment it sounds as if different ways of calculating the threshold values are used

101 and 116: were nasopharyngeal and/or throat swabs collected for PCR?

Section 2.3: it is not clear what exactly was done here: were all swabs tested for both measles and rubella or were only swabs from IgM positive patients tested for the respective virus?

Table 1: please replace „gender“ by „sex“ in the table and whole article since you talk about the biological sex rather than gender identities; the age ranges are currently overlapping – please change; please change the denominator for measles since only 7428 specimens were tested throughout the table and the whole article and please adapt all numbers and percentages accordingly

180-181: districts are not shown in the table

Table 2: footnote a. says something different than section 3.1.; could you still add the missing data for 2021 (until end of the year)?

Legend figure 1: please revise the text since it is unclear („suspected IgM-positive“)

Figures 2 and 3: please translate into English and please adapt the legends since also equivocal cases are shown

Table 3: the first part of the footnote also differs from what is said in section 3.1

267: there are no measles sub-genotypes; please compare your strains to named strains of genotype B3 if you want to further characterize them

Section 3.5: this also includes the rubella genotyping results

269-271: where is the supporting evidence for rubella genotyping? Was the full-length 739 nucleotide window used for analysis? Was the clustering with genotype 1C reference strains clear? What is known about the patient? Any recent travel or contact history, etc.?

Figure 6: please use WHO measles strain nomenclature to provide information about time and location of detection of the different strains; what are the single numbers shown close to the nodes (for bootstrap values of more than 70, I would expect numbers greater than 70 on the nodes)?; please consider using different colours for the dots (red-green colour blindness); there are far less than 62 strains shown – where is the rest?

Figure 7: as mentioned above, there are no measles sub-genotypes; please compare your strains to named strains of genotype B3 if you want to further characterize them and include this information in figure 6

289: please update the WHO definition of measles/rubella elimination

302/303: what does the introduction of the combined measles and rubella vaccine has to do with lower measles case numbers? Please explain or delete

304-306: it does not make much sense to compare positivity rates since the epidemiological settings might be completely different

371: what do you mean by „vaccine-preventableB3 measles genotype“? There is no evidence that certain genotypes are not vaccine-preventable

378: please see comments above about measles sub-genotypes and change accordingly

Discussion: please discuss the reasons for the rather high number of previously vaccinated rubella IgM positives (e.g. false positive IgM results, samples collected shortly after vaccination, etc.?)

Author Response

Reviewer 1

Open Review

English language and style

( ) Extensive editing of English language and style required
(x) Moderate English changes required
( ) English language and style are fine/minor spell check required
( ) I don't feel qualified to judge about the English language and style

Yes

Can be improved

Must be improved

Not applicable

Does the introduction provide sufficient background and include all relevant references?

( )

( )

( )

( )

Are all the cited references relevant to the research?

( )

( )

( )

( )

Is the research design appropriate?

(x)

( )

( )

( )

Are the methods adequately described?

( )

(x)

( )

( )

Are the results clearly presented?

( )

(x)

( )

( )

Are the conclusions supported by the results?

(x)

( )

( )

( )

Comments and Suggestions for Authors

The article under review presents long-term measles surveillance data from Senegal. While the information is of interest, not everything is clear and some improvements for data presentation and interpretation can be made.

Thank you very much for the interest you have shown for this article and for your insightful comments and suggestions which will undoubtedly help to improve the quality of this paper. Please see our responses in blue type and excerpts of our corrections copied below. Changes to the text are shown in red type throughout.

As general comments,

Introduction and Discussion could be more concise and focussed

Thank you for this feedback. We have improved the structure of both the introduction and discussion, as well as the concision of language. We have also re-focused both sections to better reflect the relevance of our data and findings to wider vaccination and measles policy as mentioned below.

 the number of tables and figures could be reduced. 

We have now combined Table 2 and 3 into one table (now Table 2). We have removed Figure 1 as similar inferences can be drawn from Figures 2 and 3 and the incidence estimates are described directly in Table 2. We have moved Figure 7 to a Supplementary Material document.

Also tenses in the Methods part should not be mixed to describe what has been done. 

Thank you for highlighting this. As the methods describe those employed in an ongoing surveillance program we have maintained or changed all the descriptions to the present tense throughout – except for the description of ethics approval and the data management/statistical methods relating specifically to this analysis.

In the following, several comments and suggestions are provided, referring if not otherwise stated to lines in the PDF document.

19: why were not all samples tested? Maybe leave the number of received specimens out from the Abstract in order not to confuse the readers

Thank you for the suggestion. We have removed the overall number of samples received from the abstract for clarity, and this now reads:

“Among 8082 laboratory-tested specimens from measles-suspected cases, serological evidence of measles and rubella infection was confirmed in 1303/8082 (16.1%) and 465/6714 (6.9%), respectively.”

We have clarified the number of sample not tested in the 1st paragraph of the main text (mostly due to not meeting sample volume or correct identification criteria; while also facing a shortage of either measles or rubella kits during some periods)

“60 and 65 eligible samples were not tested for measles and rubella respectively, due to shortage of measles/rubella kits, inadequate sample volumes or lack of proper identification.”

24/25: please use strain names instead.

In the abstract we hoped to highlight the ubiquitous finding of the B3 genotype in Senegal cases. We have removed reference to the sub-genotype in the abstract and give the full names of the strains in Figures 4 and S3.

Abstract conclusion differs from what is said earlier: rubella declined

Thank you. We have reworded this to better reflect our conclusion that although both viruses continue to circulate, the stalling of continued progress towards measles elimination is of particular concern. 

“The incidence of rubella is now low – 0.8 (95% CI 0.4-1.3) cases per million people in 2021 – whereas progress towards measles pre-elimination targets (<1.0 case per million people per year) appears to have stalled; there were 10.8 (95% CI 9.3-12.5) cases per million people in 2021. […] Our national surveillance data suggest that despite their low incidence both measles and rubella remain endemic in Senegal with a concerning stagnation in the decline of measles infections that represents a significant challenge to the goal of regional elimination.”

33: rubella is far less contagious than measles

We have edited the first paragraph of the introduction to better reflect this.

“Both viruses are primarily transmitted between humans by direct contact with the nose and throat secretions or respiratory droplets of infected persons [2]. Measles is a highly contagious systemic viral infection…”

53/54: please revise this part of the sentence: measles is not a threat to mortality and morbidity, but the cause of this

Apologies that we have not been able to find this wording. However, we have significantly re-worded this section of the introduction including clear wording on the burden of mortality of measles, and the specific risks of complications during measles infection to the individual patient.

57: please update virus classification

Thank you for highlighting. The sentence (with an updated reference) now reads:

“Rubella, another vaccine-preventable infection that can mimic the measles syndrome and is similarly transmitted, is caused by the rubella virus (RV) – an enveloped virus of the Rubivirus genus in the Matonaviridae family [8]”

59/60: eye defects missing

Thank you – this sentence now reads as follows:

“Rubella infection during pregnancy can result in death or congenital rubella syndrome (CRS) in the fetus, which may involve sensorineural hearing impairment, eye defects, congenital heart disease, microcephaly, developmental delay, bone alterations and damage to the liver and spleen [8-10].”

69: was rubella vaccination introduced with the MR campaign in 2013 or what is the situation for RCV?

Yes. Having previously used one dose of the measles-only vaccine, the 2013 campaign used the combined measles and rubella vaccine, which was then integrated into the routine immunisation schedule from 2014 in two dose schedule. Given the importance of the context of our data regarding vaccine policy with have clarified this and added more detail as below:

“Prior to 2014, children in Senegal were routinely immunized with one dose of measles-containing vaccine (MCV1) at 9-months of age through the national EPI, with coverage increasing from 57% in 2004 to 84% in 2013 [15]. In addition to routine immunisation services, nationwide vaccination campaigns were conducted in 2006 and 2010 using measles-only vaccine (target age group: 9–59 months) and in 2013 using the combined measles-rubella vaccine (MRV) (target age group: 9 months–14 years), with 102% administrative coverage of the 2013 campaign [15]. In 2014, the MRV replaced the measles-only vaccine in the EPI in a two-dose schedule at 9 and 15 months, with 87% coverage in 2021 [15, 16]”

80: is there any rubella-specific surveillance in Senegal?

There is no separate case-definition for rubella. Suspected cases are defined by the WHO measles case definition,and are tested only if they are negative or equivocal for measles. This is as per WHO guidance on incorporating indirect rubella surveillance within measles surveillance programs, particularly following the introduction of the combined measles-rubella vaccine.

If a rubella outbreak is detected, it is investigated as per measles, alongside targeted catch up vaccination campaigns. It is a limitation of our data – and similar data from many other regional surveillance platforms that there is not parallel active surveillance of rubella cases that may present differently to measles – particularly regarding case review of potential CRS cases. We have now clarified this in the introduction, methods section, and also more specifically as a limitation in the discussion.

“…Senegal has had established laboratory-based surveillance of measles since 2000 [17], including molecular epidemiological investigations to support global elimination. This facilitated the detection of a major measles outbreak during 2009-2010 that affected people across the country and of all ages, mirroring the experience of several other African countries [17, 18, 10]. Since the implementation of the combined MRV, rubella surveillance has been incorporated into this program, through testing all measles negative and indeterminate samples from suspected measles cases and active investigation of any detected potential outbreaks.”

89-91: is there a specific rubella case definition as well?

For the pragmatic reasons stated above, surveillance is based on the measles cases definition and there is no separate definition from rubella. However healthcare workers are able to send sera and swabs for any suspected cases of rubella, and in the investigation of an rubella outbreak, there would be additional active sampling of those considered to be symptomatic.

We have clarified that both measles and rubella surveillance are both based on the measles case definition:

“This retrospective analysis describes national data on all serological measles and rubella detections in samples collected from suspected measles cases[…] A suspected case of measles is defined according to WHO guidelines”

96-98: does the outbreak investigation team follow up on single confirmed cases or only if there are more than one (if yes, how many?)?

We have updated the following sentence to add this detail:

“Following confirmed case detections (usually three or more positive samples from the same district) an outbreak investigation team conducts active surveillance of close contacts to offer vaccination…”

101-103: what determines whether a swab is collected (how is this subset of patients determined?)?

There is no specific subset of patients or selection criteria for swab sampling. From 2019 the recommendations to health workers for sample collection changed to include both blood and nasopharyngeal swabs in all suspected measles cases. However, this recommendation has been variably implemented, and in the majority of cases, only blood samples are sent. However, during shortages of serology kits, nasopharyngeal swabs have also provided an alternative method of measles and rubella detection for outbreak control, strengthening the coverage of the surveillance system, but meaning that nasopharyngeal samples do not necessarily have a paired serological sample. As the results are not directly comparable and swabs have only been used since 2019, we have not included nasopharyngeal-only detections in incidence calculations.

111-114: please rephrase for clarity – at the moment it sounds as if different ways of calculating the threshold values are used

Thank you -  now re-phrased as below.

“For both measles and rubella, a laboratory-confirmed case is defined by an IgM absorbance value difference (ΔA) of more than 0.2. An equivocal case is defined by an absorbance value difference between 0.1 and 0.2 and all equivocal cases are re-tested for confirmation.”

101 and 116: were nasopharyngeal and/or throat swabs collected for PCR?

Nasopharyngeal swabs were collected for PCR analyses from a subset of patients form 2019. Nasopharyngeal swabs are the minimum recommended swab type, although some patients may also have had oropharyngeal swabs sent in addition to nasopharyngeal swabs. For simplicity, we have removed reference to ‘throat swabs’.

“This retrospective analysis describes national data on all serological measles and rubella detections in samples collected from suspected measles cases from sites throughout the fourteen regions of Senegal between January 2010 and April 2021, and nasopharyngeal swabs for molecular analyses from 2019 onwards."

(and retained the last line of the same paragraph):

“From 2019, nasopharyngeal swabs have been collected in viral transport medium (VTM) from a subset of patients with clinically suspected measles for RT-PCR confirmation and genotyping.”

Section 2.3: it is not clear what exactly was done here: were all swabs tested for both measles and rubella or were only swabs from IgM positive patients tested for the respective virus?

Thank you – we have added the following opening to this paragraph:

“All available nasopharyngeal swabs are tested by RT-PCR first for measles virus and then for rubella virus, if negative for measles. This process is independent of the serological results, where available.”

Table 1: please replace „gender“ by „sex“ in the table and whole article since you talk about the biological sex rather than gender identities;

We have reverted to the use of the word ‘sex’ throughout as requested.

the age ranges are currently overlapping – please change;

Thank you for pointing out this ambiguity – we have now clarified the age thresholds as below:

[0 to <2]

[2 to <5]

[5 to <15]

[15 to <50]

50+

please change the denominator for measles since only 7428 specimens were tested throughout the table and the whole article and please adapt all numbers and percentages accordingly

Thank you for highlighting this. We have updated and corrected all tables to both reflect the addition of all data from 2021 and to ensure that the denominators reflect the number tested in each category. We have also added footnotes to the tables to clarify this and updated the data described in the main text, throughout.

180-181: districts are not shown in the table

Thank you for highlighting this omission. We have now described complete data by region, also adding vaccination status. As described above, we have merged the tables to two, and updated the references to the tables in the main text accordingly.

Table 2: footnote a. says something different than section 3.1.; could you still add the missing data for 2021 (until end of the year)?

Yes – we have now been able to update our analysis to reflect the complete data for 2021, and have updated all figures, tables, and references to the data, in the text, accordingly. 

Legend figure 1: please revise the text since it is unclear („suspected IgM-positive“)

This table is now removed in order to reduce the number of figures. The data is described in the tables.

Figures 2 and 3: please translate into English and please adapt the legends since also equivocal cases are shown

Thanks for pointing this out – we have now translated the both the legends and axes, and reduced the labelling along the x axis to improve legibility.

Table 3: the first part of the footnote also differs from what is said in section 3.1

Thank you – we have updated the both table footnotes to be aligned with the text. We have indicated the number tested for each virus in the header of the tables more clearly.

267: there are no measles sub-genotypes; please compare your strains to named strains of genotype B3 if you want to further characterize them

Section 3.5: this also includes the rubella genotyping results

The strain names have been more accurately annotated in the phylogenetic trees and we have now moved the second phylogenetic tree to the supplementary material as this was a secondary analysis to the main objective of establish the circulating genotypes.

Based on deeper analysis of the gene encoding hemaglutinin, sub-genotypes were identified within the B3 genotype, the most transmissible genotype in measles, as described by other authors: Zhou, N.; Li, M.; Huang, Y.; Zhou, L.; Wang, B. Genetic Characterizations and Molecular Evolution of the Measles Virus Genotype B3’s Hemagglutinin (H) Gene in the Elimination Era. Viruses 2021, 13,1970. https://doi.org/10.3390/v13101970).

In our previous description of measles strains circulating in Senegal between 2009 and 2012 - Dia N, Fall A, Ka R, Fall A, Kiori DE, Goudiaby DG, Fall AD, Faye el HA, Dosseh A, Ndiaye K, Diop OM, Niang MN. Epidemiology and genetic characterization of measles strains in Senegal, 2004-2013. PLoS One. 2015 May 22;10(5):e0121704. doi: 10.1371/journal.pone.0121704 – we have previously noted the co-circulation of sub-genotypes B3.1 and B3.3. As shown in the Supplementary Figure S3, we used reference strains for each subgenotype to measure the proximities with the Senegalese strains.

269-271: where is the supporting evidence for rubella genotyping? Was the full-length 739 nucleotide window used for analysis? Was the clustering with genotype 1C reference strains clear? What is known about the patient? Any recent travel or contact history, etc.?

Yes, we were able to get successful amplification of the 739 discriminating nucleotides with the single respiratory sample, and using the blast tool we identified the genotype to which it belonged. The patient was a boy of 11 months old leaving in Thies district without any travel history. His symptoms were documented as a fever, rhinitis, conjunctivitis and cutaneous rash. We acknowledge that as further rubella positive respiratory samples are collected going forward, a more detailed description of any subsequent outbreak investigations including patient and contact histories will be useful for informing future rubella elimination efforts. However, we had felt this was beyond the scope of this paper.

Figure 6: please use WHO measles strain nomenclature to provide information about time and location of detection of the different strains;

Thank you we have updated Figure 6 (now Figure 4) and S3, to the full WHO nomenclature to describe the strains identified.

what are the single numbers shown close to the nodes (for bootstrap values of more than 70, I would expect numbers greater than 70 on the nodes)?;

Yes, these are bootstrap values and we have labelled values greater than 70.

please consider using different colours for the dots (red-green colour blindness);

Thank you for this suggestion. We have now labelled the Senegalese strain names using blue text instead.

there are far less than 62 strains shown – where is the rest?

We did not include any identical strains from Senegal in the phylogenetic tree in order to more clearly show the range of unique strains, which is the message we wanted to convey, rather than looking at clustering. We have updated the footnote clarify this.

Figure 7: as mentioned above, there are no measles sub-genotypes; please compare your strains to named strains of genotype B3 if you want to further characterize them and include this information in figure 6

Thank you- please see responses above.

289: please update the WHO definition of measles/rubella elimination

Yes we agree this wasn’t clear. We have clarified throughout the text that as part of the regional strategy to progress towards the 2020 elimination goal, a pre-elimination target of <1.0 case/million/year was set but we of course acknowledge that this itself is not the criteria for elimination.

302/303: what does the introduction of the combined measles and rubella vaccine has to do with lower measles case numbers? Please explain or delete

At the same time as the combined vaccine was introduced the routine schedule was stepped up from a one-dose to a two-dose regimen and there was a large campaign in 2013 prior to initiating the new schedule in 2014. However, we acknowledge this wasn’t at all clear, and have significantly re-worded sections of both the introduction (see above) and discussion to frame our findings in the context of more detailed information about changes in vaccine policy.

304-306: it does not make much sense to compare positivity rates since the epidemiological settings might be completely different

We agree and we have now removed the discussion points around positivity, particularly when comparing different settings. We have instead focused on the comparison of incidence estimates when considering temporal and region trends.

371: what do you mean by „vaccine-preventableB3 measles genotype“? There is no evidence that certain genotypes are not vaccine-preventable

We agree that this does not read as intended and have edited accordingly.

378: please see comments above about measles sub-genotypes and change accordingly

Discussion: please discuss the reasons for the rather high number of previously vaccinated rubella IgM positives (e.g. false positive IgM results, samples collected shortly after vaccination, etc.?)

We thank the reviewer for this comment as it highlights that our table may have been unintentionally misleading. We do not have data on the type of vaccine received, where given. We have therefore updated the labelling of the table and descriptions in the text to clarify that this descriptor reflects measles vaccination status (either in the form of the MCV or MRV). A proportion of rubella cases who have received a vaccination previously may not have received the MRV, as this was implemented halfway through the study period. We have therefore also added additional detail on this in the text: that when restricting to cases that occurred from 2015 or 2016 onwards, very few cases had a vaccination history:

“Only 10 (10.1%) and 7 (8.6%) IgM-positive rubella cases occurring from 2015 and 2016 onwards respectively (ie. following introduction of the combined MRV) had a documented measles vaccination history, two of whom were >10 years old.”

Reviewer 2 Report

This paper documents the prevalence of measles and rubella in Senegal from 2010 to 2021 based on laboratory diagnosis-based surveillance. Molecular biological analysis of the virus has also been conducted since 2019. The authors note that progress has been made in eliminating measles and rubella over the past 12 years, but that to achieve elimination, increased immunization coverage, maintenance of high-quality surveillance, and appropriate responses to imported cases are needed.

Although the paper needs to be improved in figures, tables, and writing style, the reviewer is considering that the paper is significant as information and as an archive for Senegal and other African countries to achieve measles and rubella elimination in the future.

Comment.

There are a few places where the descriptions are clunky. They need to be corrected.

P2, L68-L74 

The vaccine used in the regular vaccination (1st dose) is described as MCV1, but it is not clear whether the Measles-Rubella vaccine or the Measles vaccine is used. The type of vaccine is important information and should be clearly stated.

Vaccine campaigns (2nd dose) were conducted in 2006, 2010, and 2013. Whether the campaigns contributed to the decrease in the number of measles and rubella cases should be discussed and added in the text. Also, the measles-rubella vaccine was used in 2013, but the number of rubella cases reported in 2014 is higher. Discuss this as well and add in the text.

Please clearly state in the text whether the cases that were equivocal as a result of IgM antibody testing were added to the number of patients or were considered DISCARD cases

Figs. 4 and 5 show maps classified by incidence rate per million population over a 12-year period, but the locations of the outbreaks varied from year to year and the number of cases also varied. The reviewer thinks it is of little significance to show the results for the 12-year period as a whole.

In addition, when considering the geographic distribution of case, it did not take into account the impact of outbreaks occurred in neighboring countries. The impact of outbreaks in bordering countries should also be considered.

Other comments

Introduction

1) P1, L34~L36 & L57; The description method of virus name, genus and family is different from the method prescribed in the paper. Please check and rewrite.

2) P2 L56 prevenTabledisese  → preventable disease?

M&M

3)      P3, L 23; 2.4 2.3 describes real-time PCR, but there is no description of conventional RT-PCR for genotyping. Please add the description appropriately.

4)      P3, L125;  nucleotide position 1233-1682 → nucleotide position (nts) 1233-1682

5)      P3, L134 ;  Phelogenetic  Phylogenetic

6)      P3, L139; Measles Nt surveillance databases (MeaNS)→ Measles Nucleotide Surveillance. online database (MeaNS)

Results

7)      P5 L185  96.9%   This number is shown as 97.0 in Table 1.

8)      P5 L186  77.9%   This number is shown as 78.0 in Table 1.

9)      P5 L186  3% → 3.0% (Uniform number of decimal places

10)   P5 L187   2%→ 22.1% (Uniform number of decimal places

12) P6 Fig 1 Bar chart is easier to understand than Line chart. There is no point in connecting the number of patients in each year with a line.

13) P7 Fig. 2 and Fig. 3 are written in French in the legend and the title of the vertical axis. Also, the numbers on the horizontal axis of the graph are too detailed to read. It would be better to list them by year instead of by month.

14) P8 Table 3 No. of suspected cases is missing in some regions.

15) P8 Fig 4, 5 Region names are not legible in dark-colored areas.

16)   P9 L226  Futhermore → Furthermore

17)   P10, Fig 6&7;  The nomenclature of measles virus and rubella virus sequences is defined by WHO, which is easy to understand because it describes the place and week of specimen collection. The nucleotide sequence names in Fig 6 and Fig 7 should be described according to the WHO rules.

18)   P10 and Fig. 6 should also include the name of the reference strain of the 

genotype.

19)   Sequence information should be registered in DNA data banks such as Genbank,

and Accession no. should be listed.

20)   P12 L344 1:1.2?

21)   P12 L346, 1:1.4?

Discussion

22)   P11 L289  Not all six WHO regions use "<1.0 cases per million people per year" as a target for measles / rubella elimination. Please state appropriately.

Author Response

Reviewer 2

Open Review

English language and style

( ) Extensive editing of English language and style required
(x) Moderate English changes required
( ) English language and style are fine/minor spell check required
( ) I don't feel qualified to judge about the English language and style

Yes

Can be improved

Must be improved

Not applicable

Does the introduction provide sufficient background and include all relevant references?

(x)

( )

( )

( )

Are all the cited references relevant to the research?

( )

(x)

( )

( )

Is the research design appropriate?

( )

(x)

( )

( )

Are the methods adequately described?

( )

( )

(x)

( )

Are the results clearly presented?

( )

( )

(x)

( )

Are the conclusions supported by the results?

( )

(x)

( )

( )

Comments and Suggestions for Authors

This paper documents the prevalence of measles and rubella in Senegal from 2010 to 2021 based on laboratory diagnosis-based surveillance. Molecular biological analysis of the virus has also been conducted since 2019. The authors note that progress has been made in eliminating measles and rubella over the past 12 years, but that to achieve elimination, increased immunization coverage, maintenance of high-quality surveillance, and appropriate responses to imported cases are needed.

Although the paper needs to be improved in figures, tables, and writing style, the reviewer is considering that the paper is significant as information and as an archive for Senegal and other African countries to achieve measles and rubella elimination in the future.

Thank you very much your time and interest in this article, and for the very useful comments which will help us to strengthen the paper. Please see our responses in blue type and excerpts of our corrections copied below. Changes to the text are shown in red type throughout.

Comment.

There are a few places where the descriptions are clunky. They need to be corrected.

Thank you for this feedback. After some significant revisions throughout all sections of the paper, we hope that we have improved the clarity and concision of writing.

P2, L68-L74 

The vaccine used in the regular vaccination (1st dose) is described as MCV1, but it is not clear whether the Measles-Rubella vaccine or the Measles vaccine is used. The type of vaccine is important information and should be clearly stated.

Thank you – and we agree that the description of both current and historical vaccine policy was not clear and is essential context for which to interpret the data. We have therefore both clarified our previous statement and added the detail below: 

“Prior to 2014, children in Senegal were routinely immunized with one dose of measles-containing vaccine (MCV1) at 9-months of age through the national EPI, with coverage increasing from 57% in 2004 to 84% in 2013 [15]. In addition to routine immunisation services, nationwide vaccination campaigns were conducted in 2006 and 2010 using measles-only vaccine (target age group: 9–59 months) and in 2013 using the combined measles-rubella vaccine (MRV) (target age group: 9 months–14 years), with 102% administrative coverage of the 2013 campaign [15]. In 2014, the MRV replaced the measles-only vaccine in the EPI in a two-dose schedule at 9 and 15 months, with 87% coverage in 2021 [15, 16]”

Vaccine campaigns (2nd dose) were conducted in 2006, 2010, and 2013. Whether the campaigns contributed to the decrease in the number of measles and rubella cases should be discussed and added in the text.

Thank you, we have now significantly re-worded sections of both the introduction (see above) and discussion to frame our findings in the context of more detailed information about changes in vaccine policy, including the following reflections for measles:

“Our data captures trends during more than a decade of measles elimination efforts in the African region. Indeed, the large decrease in measles incidence between 2010 and 2011 may reflect the success of the 2010 vaccination campaign; the effect of the 2013 campaign on measles control is less apparent, albeit starting from a much lower baseline incidence.”

And

“….Most concerningly, there have consistently been more than 10 cases per million people annually for the past three years – with incidence previously 1.0 (95% CI 0.5-1.6) and 0.4 (95% CI 0.2-0.9) per million people per year in 2017 and 2018, respectively. This is despite stepping up to a two-dose vaccine schedule in 2014 with high population coverage and reflects similar setbacks experienced in measles control documented globally during this time period. Importantly, we did not have complete information on residential status or travel history, meaning we cannot assess the degree to which imported cases and outbreaks in surrounding countries are driving the observed incidence in Senegal.”

Also, the measles-rubella vaccine was used in 2013, but the number of rubella cases reported in 2014 is higher. Discuss this as well and add in the text.

Thank you – as described above, we hope that the text is now clear that although the combined MRV was used for the 2013 vaccine campaign, the MRV was not introduced into the routine schedule until 2014. However despite this, the administrative coverage of the 2013 campaign was reported to be high and we do not therefore have a clear explanation on the rise in cases in 2014. The lack of data on travel and contact history, and on the specific type of vaccine previously received by each suspected case limits our ability to further assess this and we have now made these particular limitations clearer in the text.

Please clearly state in the text whether the cases that were equivocal as a result of IgM antibody testing were added to the number of patients or were considered DISCARD cases?

Although we have reported both the IgM confirmed and equivocal results in all the tables for clarity we have now clearly emphasized in the methods that incidence calculations were based on IgM positive cases only, and disregarded equivocal cases:

“National incidence rates of laboratory-confirmed measles and rubella were estimated per million people (all ages) per year, based on IgM-positive detections (i.e. not including cases with equivocal levels of IgM on repeated testing)”

Figs. 4 and 5 show maps classified by incidence rate per million population over a 12-year period, but the locations of the outbreaks varied from year to year and the number of cases also varied. The reviewer thinks it is of little significance to show the results for the 12-year period as a whole.

Thank for you for this reflection. While updating all the analyses to now include complete data for 2021 we were also able to conduct new incidence estimates by region at different time periods. We have now updated Figures 4 and 5 to show three different time periods during the study for each infection. We agree that the data shows a heterogenous picture of outbreaks and incidence and it is precisely this which we hope to display in the article. Although the individual maps are now smaller, we hope that this updated version combines both regional and temporal estimates in a manner that is more informative overall. For contrast we have included the previous version (updated with 2021 data) in the supplementary material, which could be included or not depending on the reviewers feedback and editors discretion.

In addition, when considering the geographic distribution of case, it did not take into account the impact of outbreaks occurred in neighboring countries. The impact of outbreaks in bordering countries should also be considered.

We think it is highly likely that outbreaks in and imported cases in neighbouring countries play a significant role in variable incidence and difficulty progressing towards complete elimination. However, we do not have comprehensive data from other countries, extensive consideration of this may be beyond the scope of what we are able to infer from our data in this paper. We have now noted this as an important limitation in the discussion:

“Importantly, we did not have complete information on residential status or travel history, meaning we cannot assess the degree to which imported cases and outbreaks in surrounding countries are driving the observed incidence in Senegal.”

Other comments

Introduction

  • P1, L34~L36 & L57; The description method of virus name, genus and family is different from the method prescribed in the paper. Please check and rewrite.

We have updated the family name of the rubella virus which was out of date: “…Rubivirus genus in the Matonaviridae family”

2) P2 L56 prevenTabledisese  → preventable disease?

Thank you – corrected

M&M

  • P3, L 23; 2.4 2.3 describes real-time PCR, but there is no description of conventional RT-PCR for genotyping. Please add the description appropriately.

Thank you we have now added the full details as follows:

“All RT-PCR measles or rubella positive nasopharyngeal samples are genotyped. After an initial step of cDNA synthesis using the RevertAid First Strand cDNA Synthesis Kit (Thermo Scientific, Lithuania), a nested PCR is performed to amplify targeted fragments for both measles and rubella viruses. For measles virus, PCR is carried out using primers previously described (17). The GoTaq® DNA Polymerase (Promega, USA) is used for amplifications. The first PCR is carried out in a total reaction volume of 27.25 μl containing 12 μl of nuclease free water, 5 μl of 5X GoTaq® Reaction Buffer, 2 μl of MgCl2, 1 μl of dNTP (10mM), 1 μl of forward primer (20 μM), 1 μl of reverse primer (20 μM), 0.25 μl of GoTaq® DNA Polymerase (5u/µl) and 5 μl of cDNA template. The reaction mixture is amplified in a thermocycler under the following conditions: an initial denaturation step of 3 min at 95°C followed by 35 PCR cycles at 95°C for 30 sec, 55°C for 30 sec, and 72°C for 1 min followed by a final step at 72°C for 5 min. The Nested PCR is performed on the resulting diluted (1:50) amplicon using the same condition as the first PCR. Regarding amplification of the rubella virus, the reaction mixture of the first PCR contains 5 μl of cDNA template, 5 μl of 5X GoTaq® Reaction Buffer, 1 μl of RubF1 (5’CCCACCGACACCGTGATGA 3’) forward primer (20 μM), 1 μl of RubR1 (5’CCAGGTCTGCCGGGTCTC 3’) reverse primer (20 μM), 2 μl of MgCl2, 1 μl of dNTP (10mM), 0.25 μl of GoTaq® DNA Polymerase (5u/µl) and 12 μl nuclease free water and the cycling conditions are 3 min at 95°C followed by 35 cycles at 95°C for 30 sec, 55°C for 30 sec, and 72°C for 1 min, with a final step at 72°C for 5 min. For the nested PCR, internal primers RubF2 (5’-GTGATGAGCGTGTTCGCCC-3’) and RubR2 (5′-GCDGTGGTGTGTGTGCC-3 ‘), and 1 μl of the first PCR product are used. Amplification products are analyzed in a 1.5 % agarose gel stained with ethidium bromide, using 1xTAE as the electrophoresis running buffer.”

4)      P3, L125;  nucleotide position 1233-1682 → nucleotide position (nts) 1233-1682

Thank you – now amended

5)      P3, L134 ;  Phelogenetic → Phylogenetic

Thank you – now corrected

6)      P3, L139; Measles Nt surveillance databases (MeaNS)→ Measles Nucleotide Surveillance. online database (MeaNS)

Thank you – now updated as advised.

Results

Based on our updated analysis incorporating complete 2021 data, we have updated all the tables and checked and corrected all values and percentages. We have also now merged the tables to reduce the overlap in data being presented. We therefore now only report prior vaccination (and not the reciprocal unvaccinated values) in Table 2 to incorporate both regional and annual incidence estimates, alongside regional values for vaccination status. We have ensured that the text and tables are now aligned with consistent numbers of decimal places.

7)      P5 L185  96.9%   This number is shown as 97.0 in Table 1.]

Corrected as above

8)      P5 L186  77.9%   This number is shown as 78.0 in Table 1.

Corrected as above

9)      P5 L186  3% → 3.0% (Uniform number of decimal places)

Thank you – amended as above

10)   P5 L187   2%→ 22.1% (Uniform number of decimal places)

Thank you – amended as above

12) P6 Fig 1 Bar chart is easier to understand than Line chart. There is no point in connecting the number of patients in each year with a line.

Thank you for this feedback, in an attempt to reduce the number of figures and tables, we have now removed this figure, as the annual incidence data is already described in Table 2.

13) P7 Fig. 2 and Fig. 3 are written in French in the legend and the title of the vertical axis. Also, the numbers on the horizontal axis of the graph are too detailed to read. It would be better to list them by year instead of by month.

Thank for pointing out the translation error – which is now corrected as well as updating the axes with your suggestion, indicating January of each year.

14) P8 Table 3 No. of suspected cases is missing in some regions.

Thank you for highlighting this omission. Complete data for each region is now described in Table 2.

15) P8 Fig 4, 5 Region names are not legible in dark-colored areas.

Thank you – we were not able to change the colour of the region name for Kédougou only, the darkest region, and lightening all labels proved illegible elsewhere. However, we suggest that given the new changes to the figure, the name of the darker regions are now visible in the adjacent maps where the colouring is lighter. If required we could also move the regional name for Kédougou specifically to appear outside of the map, adjacent to the region, but felt the overall visual appearance was best in the version we have provided. The larger versions of these maps for which the regional names are easier to read are in included in the supplementary material – pending feedback on whether the maps at different time periods are preferred.

16)   P9 L226  Futhermore → Furthermore

Thank you - amended

17)   P10, Fig 6&7; The nomenclature of measles virus and rubella virus sequences is defined by WHO, which is easy to understand because it describes the place and week of specimen collection. The nucleotide sequence names in Fig 6 and Fig 7 should be described according to the WHO rules.

We have now updated the phylogenetic trees to include full WHO nomenclature to describe the measles virus sequences.

18)   P10 and Fig. 6 should also include the name of the reference strain of the genotype. Corrected.

19)   Sequence information should be registered in DNA data banks such as Genbank, and Accession no. should be listed. Indeed, it is a good suggestion, sequences submission will be done in a short time.

20)   P12 L344 1:1.2?

Thanks for pointing these out. We have updated these sentences as follows:

Our reported male:female ratio of 1:1.2 in IgM-positive measles cases is consistent with findings from Zaidi et al [30], and disproportionate to child sex ratio (<7years) of 1:1.06 in Senegal [www.unfpa.org]. The even higher male:female ratio of suspected cases of 1:1.4 may suggest a difference in care-seeking or other factors influencing case-finding, as sex-related differences in measles incidence is not consistently reported elsewhere [9].

21)   P12 L346, 1:1.4?

Amended as above

Discussion

22)   P11 L289  Not all six WHO regions use "<1.0 cases per million people per year" as a target for measles / rubella elimination. Please state appropriately.

Yes we agree this wasn’t clear. We have clarified throughout the text that as part of the African regional strategy to progress towards the 2020 elimination goal, a pre-elimination target of <1.0 case/million/year was set but we of course acknowledge that this itself is not the criteria for elimination.